# *Smed-pou4-2* regulates mechanosensory neuron regeneration and function in planarians

Ryan A McCubbin[1†], Mohammad A Auwal[1†], Shengzhou Wang[1], Sarai Alvarez Zepeda[1], Roman Sasik[2], Robert W Zeller[1], Kelly G Ross[1], Ricardo M Zayas[1]*

[1]Department of Biology, San Diego State University, San Diego, United States; [2]Center for Computational Biology and Bioinformatics, University of California, San Diego, San Diego, United States

*For correspondence:
rzayas@sdsu.edu

†These authors contributed equally to this work

Competing interest: The authors declare that no competing interests exist.

## eLife Assessment

This is a **valuable** study that explores the role of the conserved transcription factor POU4-2 in the maintenance, regeneration, and function of planarian mechanosensory neurons. The authors present **convincing** evidence provided by gene expression and functional studies to demonstrate that POU4-2 is required for the maintenance and regeneration of mechanosensory neurons and mechanosensory function in planarians. Furthermore, the authors identify conserved genes associated with human auditory and rheosensory neurons as potential targets of this transcription factor.

**Abstract** POU4 homologs are involved in the development of sensory cell types across diverse species, including cnidarians, ascidians, and mammals. Whether these developmental regulators are redeployed during adult tissue maintenance and regeneration remains an open question in regenerative biology. Here, we investigated the role of the *Schmidtea mediterranea* BRN3/POU4 homolog, *Smed-pou4-2* (*pou4-2*), in the regeneration of mechanosensory neurons. We found that *pou4-2* is regulated by the SoxB1 homolog *soxB1-2* and is expressed in a distinct population of ciliated sensory cells that detect water flow. Transcriptomic analysis of *pou4-2*-deficient planarians revealed enrichment for conserved genes associated with human auditory and vestibular function, suggesting that planarian rheosensory neurons share molecular features with mammalian inner ear hair cells. Expression of these conserved genes was significantly reduced following RNAi-mediated knockdown of *pou4-2*. To determine whether these transcriptional changes had functional consequences, we assessed the impact of *pou4-2* knockdown on sensory function. *pou4-2* RNAi resulted in impaired mechanosensation in both uninjured and regenerating planarians. Together with the loss of terminal differentiation markers in mechanosensory neurons, these findings identify *Smed-pou4-2* as a key regulator of mechanosensory neuron identity in planarians and support the idea that conserved sensory specification programs are redeployed during adult tissue regeneration.

## Introduction

Most animals, including mammals, have a limited capacity for neuronal regeneration. In contrast, organisms like fish and salamanders can effectively regenerate neurons, and some invertebrates are capable of dramatic whole-body regeneration. The freshwater planarian *Schmidtea mediterranea* is among a handful of research organisms capable of restoring virtually any lost or damaged tissue and can regenerate entire animals from small body fragments (***Goldstein and Srivastava, 2022***;

*Ivankovic et al., 2019*). *S. mediterranea* possesses a population of adult pluripotent stem cells called neoblasts, which proliferate and differentiate to replace all missing tissues (*Baguñà, 2012*; *Newmark and Sánchez Alvarado, 2002*; *Reddien, 2018*). This stem cell population is postulated to include a heterogeneous pluripotent pool poised to acquire lineage-specific cell fates as needed (*Raz et al., 2021*). One of the extraordinary properties of planarians is the capacity for constant neuronal turn-over and regeneration of neuronal cell types, many of which are conserved with vertebrates (*Brown and Pearson, 2017*; *Ross et al., 2017*). Despite recent advances, much remains unknown about the molecular basis of neurogenesis and the signals that regulate neuronal turnover (*Lee, 2023*). Previous studies found that *soxB1-2*, a mammalian Sox1/2/3 homolog, regulates the regeneration of ectodermal cell type subsets in planarians, including many uncharacterized sensory neurons (*Ross et al., 2018*). One prominent population of *soxB1-2*-regulated sensory cells, organized in a striking dorsal stripe pattern, functions in mechanosensation and is marked by *polycystic kidney disease-like* homologs (*Ross et al., 2024*; *Ross et al., 2018*; *Figure 1A*). In this study, we focused on identifying mechanisms downstream of *soxB1-2* that contribute to the specification of mechanosensory cells in the dorsal ciliated stripe.

POU transcription factor family genes play key roles in the development and function of many neuronal subtypes. To date, dozens of POU genes have been identified in vertebrates and invertebrates, and their roles in the differentiation and survival of diverse neuronal subtypes have been characterized (*Leyva-Díaz et al., 2020*). In many species, Brn3/POU4 transcription factors play important roles in specifying and maintaining the identities of various cell populations in the developing peripheral sensory nervous system. A notable example is *Nematostella vectensis NvPOU4*, which is required to maintain and differentiate cnidocytes, a population of mechanosensing cells exclusive to the phylum Cnidaria (*Tournière et al., 2020*). The homologous role of *NvPOU4* in cnidarians suggests that the functional role of *pou4* is ancient and conserved across distantly related phyla. Additionally, *pou4* is part of a proneural regulatory cascade that produces epidermal sensory neurons in *Ciona intestinalis*; induction of ectopic *pou4* expression in the developing epidermis of *Ciona* larvae converts epidermal cells to sensory neurons, resulting in a striking hyper-ciliated phenotype (*Chen et al., 2011*).

In mice, *Pou4f3* is expressed in the inner ear sensory epithelia during embryonic development and is required for the survival of vestibular hair cells of the auditory system (*Erkman et al., 1996*; *Xiang et al., 1997*) - its targeted deletion results in impaired hearing and balance. Hair cells of the inner ear are crosslinked by stereocilia on their apical ends that function as mechanosensors, converting vibration-induced mechanical force into signals carried by auditory nerve fibers to the central nervous system (*Goutman et al., 2015*). Although a small number of hair cells differentiate in $Pou4f3^{-/-}$ mice, their failure to form stereociliary bundles leads to apoptosis (*Xiang et al., 1998*). Thus, *Pou4* has conserved roles in the differentiation, maintenance, and survival of ciliated mechanosensory neurons. Unlike birds and fish, mammals lack the ability to regenerate hair cells after they are lost, resulting in permanent deafness (*Edge and Chen, 2008*). Recent studies show that POU4 can be used as a reprogramming co-factor to restore hair cells in mammals (*Chen et al., 2021*; *Iyer et al., 2022*). However, whether developmental regulators like *POU4* play similar roles in adult tissue maintenance and regeneration remains to be fully resolved (*Seifert et al., 2023*).

In *S. mediterranea*, a search for candidate planarian OCT4 homologs, a gatekeeper of pluripotency also known as POU5F1 in humans (*Zeineddine et al., 2014*), revealed six genes containing a POU-specific domain and a POU-homeodomain, and the two genes most similar to hPOU4F3 were named *Smed-pou4-1* and *Smed-pou4-2* (*Onal et al., 2012*; referred to as *pou4-1* and *pou4-2* hereon). *pou4-1* (also referred to as *pou4-like and pou4-like-1*) was identified as downstream of COE (*Cowles et al., 2014*), a transcription factor required for neurogenesis widely conserved across metazoans (*Demilly et al., 2011*), and is responsible for maintaining proper neuronal architecture in the cephalic ganglia as well as photoreceptor pigmentation (*Cowles et al., 2014*). More recently, our lab and others observed robust sensory defects in *pou4-2(RNAi)* planarians (*Elliott, 2016*; *McCubbin, 2022*; *Wang, 2019*). Under normal conditions, *S. mediterranea* worms display a stereotyped behavior by shortening their bodies in response to vibrations and water currents across their dorsal side (rheo-sensation). We found that *pou4-2(RNAi)* planarians failed to react to this sensory input, suggesting a critical role for *pou4-2* in mechanosensory neuron function (*Elliott, 2016*; *Wang, 2019*); however, its role in regenerative neurogenesis is not well understood. Here, we examined the function of *pou4-2* in mechanosensory neuron regeneration.

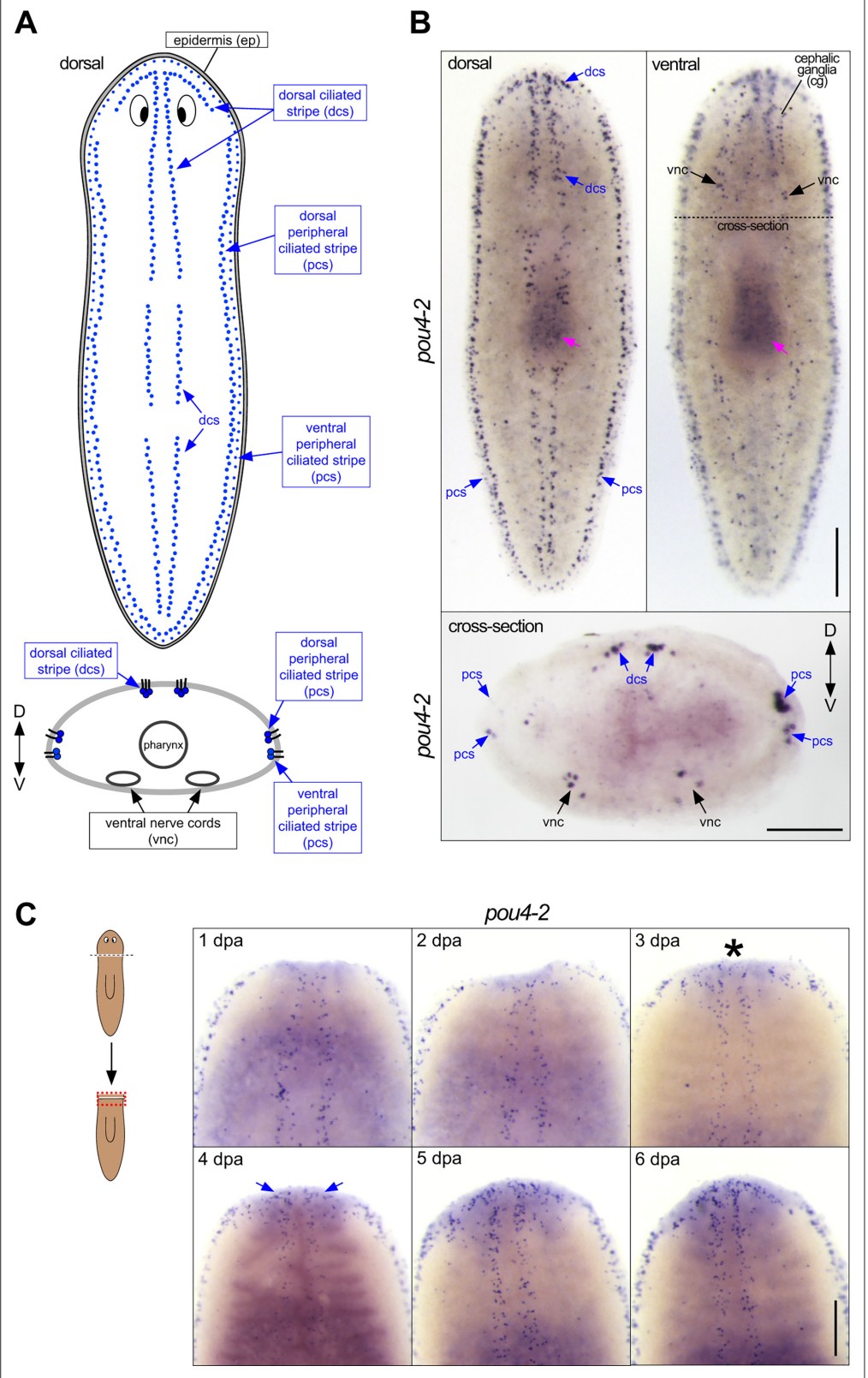

**Figure 1.** *Smed-pou4-2* is expressed in the ciliated stripes. (**A**) Schematic of dorsal and peripheral sensory cell patterns implicated in mechanosensation. (**B**) Whole-mount in situ hybridization (WISH) of *pou4-2* reveals stereotyped mechanosensory neuron expression in the dorsal head tip, body periphery, dorsal ciliated stripe (dcs), and ventral nerve cords (vnc). The dashed line indicates the cross-section plane shown below. Scale bar =

*Figure 1 continued on next page*

*Figure 1 continued*

200 µm. (**C**) Regeneration time course showing reappearance of *pou4-2* expression in the blastema beginning at day 3 post-amputation (asterisk), with re-establishment of dorsal stripe expression by day 7. Blue arrows mark the reappearing dorsal ciliated stripe pattern. Anterior is up. Scale bars = 200 µm; n ≥3.

The online version of this article includes the following figure supplement(s) for figure 1:

**Figure supplement 1.** Expression of *Smed-pou4-2* during development of *S. mediterranea.*

---

In this study, we mapped the expression of *pou4-2* and assessed its function through RNAi and RNA-seq. Loss of *pou4-2* expression coincides with loss of mechanosensation, which is not restored in regenerated *pou4-2(RNAi)* planarians. Analysis of the *pou4-2⁺* cell gene expression profile uncovered that many genes regulated by Pou4-2 activity are necessary for proper mechanosensory neuron function and are homologs of human genes involved in hair cell function and auditory perception. In many organisms, the proneural *atonal* genes function in the same gene regulatory network as *pou4* (*Leyva-Díaz et al., 2020*). However, this relationship does not appear to be conserved in planarians. Our findings suggest that *pou4-2* participates in a regulatory cascade involved in the specification of distinct sensory neuron populations. This study demonstrates that *pou4-2* plays a key regulatory role in the differentiation, maintenance, and regeneration of ciliated mechanosensory neurons in planarians.

## Results

### *Smed-pou4-2* is expressed in planarian mechanosensory neurons

Our previous work demonstrated a key role for *soxB1-2* in the differentiation and function of sensory neuron subclasses in the planarian *Schmidtea mediterranea* (*Ross et al., 2018*). A subset of *soxB1-2*-regulated genes is abundantly expressed in a discrete pattern called the dorsal and peripheral ciliated stripes (*Figure 1A*), which contain ciliated sensory neurons involved in detecting water flow (rheosensation). We took a candidate-based approach to gain mechanistic insight into how the sensory stripe cells are specified from a heterogeneous *soxB1–2⁺* progenitor pool. POU4 genes are involved in the development of sensory organs detecting mechanical stimulation in divergent organisms (*Manley and Ladher, 2008*; *Zhao et al., 2020*). Thus, we investigated the expression and function of *S. mediterranea pou4* genes. The planarian genome encodes two POU4 homologs, *pou4-1* (also referred to as *pou4-like*) and *pou4-2* (*Onal et al., 2012*). In previous work, we found that *pou4-1* is expressed in the planarian CNS (*Cowles et al., 2014*). In contrast, analysis of *pou4-2* using whole-mount in situ hybridization (WISH) showed expression localized in the dorsal head tip and dorsal and peripheral ciliated stripes of intact planarians (*Figure 1A–B*), a stereotyped pattern common to rheosensory genes (*Ross et al., 2018*). In addition, *pou4-2* expression was detected in cells dispersed throughout the body in a subepidermal punctate pattern and in the cephalic ganglia and ventral nerve cords (*Figure 1B*). Because POU4 genes have been implicated as terminal selectors in diverse organisms (*Leyva-Díaz et al., 2020*), we examined the expression pattern of *pou4-2* in regeneration blastemas to determine whether its activation coincides with late differentiation stages. During the first 24 hr of regeneration, *pou4-2* expression was absent from the blastema. We first detected clear expression on day 3 of regeneration (*Figure 1C*). The patterning of *pou4-2* expression in the blastema at day 3 was less organized and not confined to its normal spatial location, with *pou4-2⁺* cells sparsely scattered throughout the regeneration blastema. During days 4 and 5, *pou4-2⁺* cells began to repopulate the stereotypical stripe expression pattern, and by day 7, proper patterning was restored. In planarians, the regeneration blastema is populated by post-mitotic progenitors (*Reddien, 2018*). The delayed re-establishment of *pou4-2* expression suggests that it is required in the later stages of cell differentiation in regeneration. Interestingly, embryonic expression of *pou4-2* is predominantly detected during Stages 5–6, which coincide with organogenesis and nervous system development in *S. mediterranea* (*Figure 1—figure supplement 1*). While the functional role of *pou4-2* in embryogenesis remains unknown, these data suggest that its temporal dynamics are consistent with a role in late-stage differentiation, as observed in adult regeneration (*Davies et al., 2017*).

*soxB1-2* is expressed in and regulates transcription in dorsal and peripheral ciliated stripe neurons as well as in other neural populations, and the epidermis of *S. mediterranea* (*King et al., 2024*; *Ross et al., 2018*). Therefore, we searched the existing scRNA-seq data from the entire body and brain

(*Fincher et al., 2018*) to examine the potential relationship between *soxB1-2* and *pou4-2*. First, we extracted 1427 putative neuronal cells expressing *soxB1-2*. We resolved 19 distinct *soxB1-2⁺* neuronal clusters (*Supplementary file 1* and *Figure 2A*), of which cluster 8 was marked by *pou4-2*. The presence of *synapsin* and *synaptogamin* (neural markers) in cluster 8 indicated that these cells are neurons (*Figure 2B*). We combed through the dataset to identify genes that are differentially expressed in the *pou4-2⁺* cluster (*Figure 2C*; *Supplementary file 2*); highly enriched genes included *pkd1L-2* and *hmcn-1-L*, which are highly enriched in the planarian rheosensory organ (*Ross et al., 2018*). Because *pkd1L-2* and *hmcn-1-L* expression requires *soxB1-2* activity, we hypothesized that *soxB1-2* regulates *pou4-2* expression. Thus, we treated planarians with *soxB1-2* or *pou4-2* dsRNA (the RNAi treatment scheme is depicted in *Figure 2—figure supplement 1A*) and processed them for WISH. We observed a significant reduction in mechanosensory neuron-patterned *pou4-2* expression in *soxB1-2(RNAi)* planarians, whereas *pou4-2* expression in the central nervous system remained unaltered (*Figure 2D*). Conversely, *soxB1-2* expression was downregulated in mechanosensory neuron-patterned areas important for rheosensation (*Figure 2—figure supplement 1B*). However, other areas enriched with *soxB1-2* expression, such as the auricles - anteriorly positioned lateral flaps involved in chemotaxis (*Almazan et al., 2021*), the pharynx - an organ serving as the entrance and exit to the digestive system (*Ishii, 1962*), and the epidermis, were unaffected by *pou4-2* RNAi. Together, these results support a model in which *soxB1-2* positively regulates *pou4-2* expression specifically in mechanosensory neurons, without affecting *pou4-2* expression in other neural or epidermal populations.

In many organisms, *pou4* and the proneural *atonal* genes are part of the same gene regulatory network (*Leyva-Díaz et al., 2020*). In *Ciona intestinalis*, *atonal* and *pou4* are part of a regulatory cascade downstream of Notch signaling that generates sensory neurons (*Joyce Tang et al., 2013*). In mice, *atoh1* is required for differentiation of multiple mechanosensory neuron types and stimulates expression of *pou4f3* to promote hair cell fate (*Yu et al., 2021*), and overexpression of *pou4f3* together with *atoh1* and *gfi1* in mouse embryonic stem cells can induce inner ear hair cell differentiation in vitro (*Costa et al., 2015*). There are three *atonal* homologs in the planarian genome, but none appear to operate in the same regulatory network as *pou4-2*. *atoh-1* is expressed in a discrete neuronal population in the cephalic ganglia, while *atoh8-1* and *atoh8-2* are expressed in stem cells in the mesenchyme (*Cowles et al., 2013*). *pou4-2* expression was unaffected after RNAi inhibition of all *atonal* genes in regenerated planarians (*Figure 2—figure supplement 2*). Thus, the functional relationship between *pou4* and *atonal* observed in other animals does not appear to be conserved in planarians, based on current expression data and RNAi analyses.

### *Smed-pou4-2* regulates genes involved in sensory neuron terminal fate

Based on known roles of Pou4 genes, we hypothesized that *pou4-2* is required for sensory neuron differentiation and function. To test our hypothesis, we performed RNAi of *pou4-2* and pinpointed time points wherein the *pou4-2* transcripts were robustly downregulated (not shown) and subsequently performed whole-animal RNA-seq on day 12 of the RNAi knockdown for *pou4-2(RNAi)* and control animals (see Materials and methods). Analysis of the resulting data revealed downregulation of putative *pou4-2* target genes (*Supplementary file 3*). Because Pou4 genes are predicted to function as transcriptional activators, we focused further analyses on the downregulated gene set. *pou4-2* RNAi RNA-seq uncovered 72 significantly downregulated genes (FC ≥1.4, adjusted p-value <0.1; *Figure 3A* and *Supplementary file 3*). GO analysis of the *pou4-2*-downregulated gene set revealed significant enrichment in 'Mechanosensation' (*Supplementary file 4*), including previously characterized genes we assessed to have roles in planarian mechanosensory modalities like vibration sensation and rheosensation, such as the polycystic kidney disease gene homologs *pkd1L-2* and *pkd2L-1* genes (*Ross et al., 2024*; *Ross et al., 2018*). This result is consistent with prior observations of the *pou4-2* RNAi phenotype in scRNA-seq studies (*King et al., 2024*) and studies on the role of Notch signaling in planarians (*Elliott, 2016*). The discrete expression and *pou4-2* RNA-seq dataset motivated us to characterize the regulatory role of this transcription factor in planarian sensory neuron function and regeneration.

The mechanosensory neurons in the rheosensory organ are distinguished by the expression of multiple sensory neural function genes and consist of at least two distinct populations, marked by the expression of terminal markers *polycystic kidney disease 1 like-2* (*pkd1L-2*) and *hemicentin-1-like* (*hmcn-1-L*) (*Ross et al., 2018*). The evolutionarily conserved PKD1L-2 is a cation channel pore

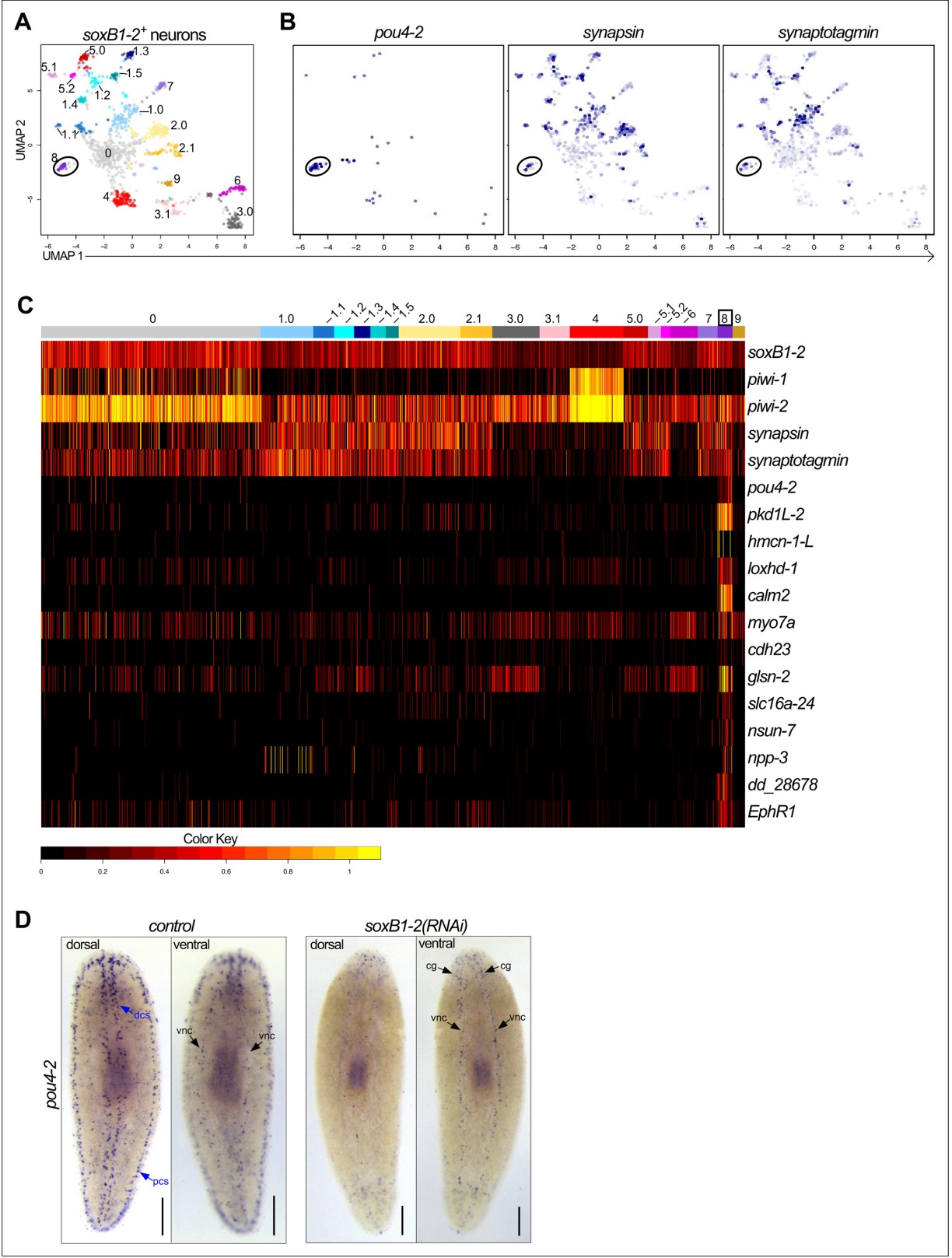

**Figure 2.** *Smed-pou4-2* is positively regulated by *soxB1-2*. (**A**) UMAP of *soxB1-2*+ neuronal subclusters from scRNA-seq data. (**B**) *pou4-2*, *synapsin,* and *synaptogamin* are enriched in cluster 8. (**C**) Heatmap of genes examined in this study demonstrates their differential expression in the *pou4-2*+ cell cluster. (**D**) *soxB1-2* RNAi reduces *pou4-2* expression in the mechanosensory dorsal and peripheral ciliated stripes (dcs and pcs) but not in the ventral nerve cords (vnc) or cephalic ganglia (cg). Scale bars = 200 µm; n ≥3 worms tested, with all samples displaying similar expression patterns.

*Figure 2 continued on next page*

*Figure 2 continued*

The online version of this article includes the following figure supplement(s) for figure 2:

**Figure supplement 1.** Schematic of RNAi treatment and reciprocal expression analysis.

**Figure supplement 2.** Knockdown of *atonal* genes does not alter *pou4-2* expression in regenerating animals.

component required for mechanosensation and cilia function (*Patel, 2015*) *pkd1L-2(RNAi)* planarians exhibit prominent rheosensory defects (*Ross et al., 2024*; *Ross et al., 2018*). On the other hand, *hmcn-1-L* is an extracellular matrix component involved in anchoring mechanosensory neurons to the epidermis (*Vogel and Hedgecock, 2001*); no detectable sensory defects were observed in *hmcn-1-L(RNAi)* planarians (*Ross et al., 2018*). Consistent with the scRNA-seq data, we found that a subset of *pou4-2*+ cells co-expressed *pkd1L-2* (74.8%) or *hmcn-1-L* (28.4%), representing two distinct mechanosensory neuron subtypes (*Figure 3B*, *Supplementary file 7*). We consistently observed variable expression levels; some cells showed high expression of *pou4-2* and low expression of terminal markers (arrows in *Figure 3B*), while others showed lower expression of *pou4-2* but high expression of terminal markers (arrowheads in *Figure 3B*), and some had high expression of *pou4-2* and the terminal markers (white dashed box in *Figure 3B*). We also observed *pou4-2*+ cells lacking detectable expression of either *pkd1L-2* or *hmcn-1-L*, which may represent late-stage progenitors that have not yet initiated terminal marker expression. Thus, the variable *pou4-2* and terminal marker expression could be due to *pou4-2* transcripts initially appearing at high levels in differentiating progenitor cells (initially terminal marker negative) to activate transcription of terminal markers and then persisting at lower levels in terminally differentiated cells.

Next, we asked if the function of *pou4-2* is required to maintain and regenerate *pkd1L-2* and *hmcn-1-L* expression in the dorsal and peripheral ciliated stripes. We conducted WISH on intact and regenerated planarians treated with dsRNA over a time course (*Figure 1—figure supplement 1A*). Like *pou4-2*, in control intact or regenerated planarians, *pkd1L-2* and *hmcn-1-L* were expressed in the head tip, dorsal ciliated stripe, and dorsal and ventral peripheral stripes (controls in *Figure 3C*). In *pou4-2(RNAi)* animals, expression of both marker genes was strongly reduced, particularly in intact animals. In regenerating animals, expression was detected at lower levels and in scattered patterns. Interestingly, in the regenerates, minimal *pkd1L-2* and *hmcn-1-L* expression was observed in the regeneration blastema. We also detected lower levels of expression of *hmcn-1-L* at scattered locations near peripheral stripes that were unaffected by *pou4-2* RNAi (*Figure 3C*, dashed red boxes). These results indicate that *pou4-2* is necessary for maintaining *pkd1L-2* and *hmcn-1-L* expression in the most prominent dorsal and peripheral ciliated stripe cell populations. The persistence of *hmcn-1-L* expression in some peripheral cells may reflect partial knockdown efficiency or that *pou4-2* function is not required in all *hmcn-1-L*+ cells.

To test whether the population of *pou4-2*+/terminal marker⁻ cells in dorsal ciliated stripe constitutes sensory neuron progenitors that have yet to express terminal markers in support of our hypothesis above, wild-type planarians were X-ray-treated with ~100 Gy, a dose reported to deplete early progenitors after 24 hr and late progenitors within 7 days (*Eisenhoffer et al., 2008*). We performed WISH analysis at 0, 3, 4, 4.5, 5, and 5.5 days post-irradiation (dpi). We quantified *pou4-2*+ cells lacking expression of *pkd1L-2* and *hmcn-1-L*, compared to *pou4-2*+ cells co-labeled using a single fluorophore mix of *pkd1L-2* and *hmcn-1-L* riboprobes (*Figure 3—figure supplement 1A*). By 5.5 dpi, we noticed an obvious decrease in *pou4-2*+/terminal marker⁻ compared to the number of *pou4-2*+/terminal marker⁺ cells (*Figure 3—figure supplement 1B–C*). The temporal decline in *pou4-2*+ cells closely resembled that of *agat-1*+ late progenitors, suggesting a similar position within the differentiation trajectory. In contrast, *piwi-1*+ neoblasts and *prog-1*+ early progenitors were almost entirely depleted by 2–3 days dpi, supporting the idea that *pou4-2* marks a later-stage progenitor population (*Figure 3—figure supplement 1D*). These data support a model in which *pou4-2* expression is established in late-stage progenitor cells prior to the expression of terminal sensory neuron genes and remains expressed at lower levels to maintain expression of sensory function genes in terminally differentiated mechanosensory neurons.

### *Smed-pou4-2* regulates genes implicated in ciliated cell structure organization, cell adhesion, and nervous system development

To further investigate the role of *pou4-2* in regulating the differentiation of mechanosensory neurons, we selected eight additional genes from either the *pou4-2* RNAi-downregulated gene set identified

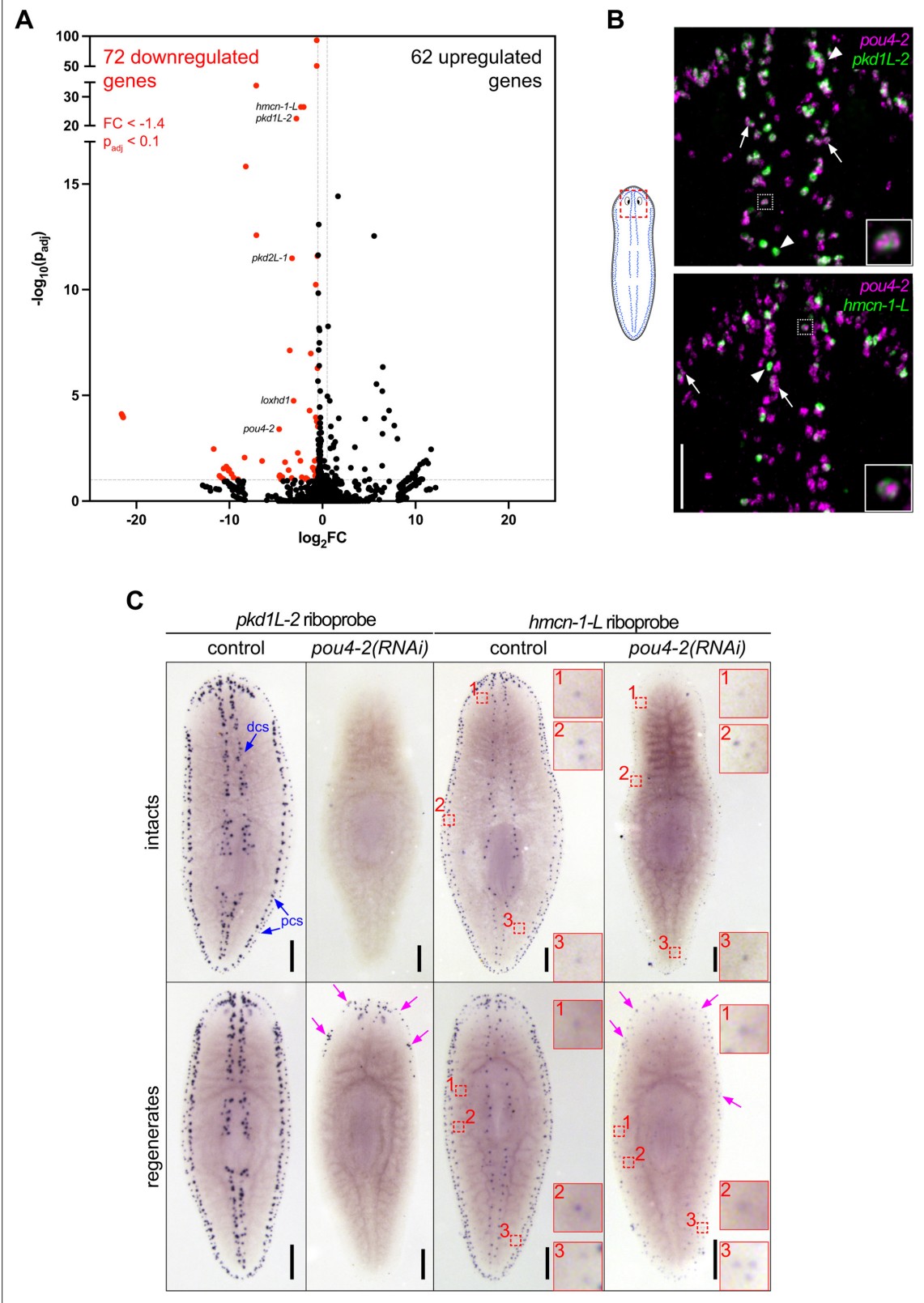

**Figure 3.** Identification of genes regulated by *Smed-pou4-2* using RNA-seq. (**A**) Volcano plot of genes differentially expressed in *pou4-2(RNAi)* animals compared to controls (FC ≥1.4, p-adj ≤0.1). A subset of genes examined in this study is highlighted on the plot and demonstrates significant downregulation. (**B**) Co-localization analysis by double-fluorescence in situ hybridization reveals that 74.8% of *pou4-2*+ cells co-express *pkd1L-2*, and 28.4% co-express *hmcn-1-L*. White boxed cells shown in insets show high *pou4-2* and terminal marker expression and are displayed at higher

*Figure 3 continued on next page*

*Figure 3 continued*

magnification. White arrowheads point to examples where terminal marker gene expression is much brighter than *pou4-2* expression. White arrows mark *pou4−2⁺* cells with high expression of *pou4-2* and low expression of the terminal marker genes. Scale bar = 100 μm. (**C**) WISH of *pkd1L-2* and *hmcn-1-L* in control and *pou4-2(RNAi)* 10 day regenerates. Terminal marker expression is strongly reduced in RNAi animals. Numbered red boxes demonstrate a population of scattered *hmcn-1-L⁺* cells that persist following *pou4-2* RNAi and are shown in corresponding zoomed-in insets. Blue arrows denote expression in the dorsal and peripheral ciliated stripes (dcs and pcs, respectively). Note that some *pkd1L-2* and *hmcn-1-L* expression was detectable in regenerates (magenta arrows). Anterior is to the top. Scale bars = 200 μm; n ≥3 worms tested, with all samples displaying similar expression patterns.

The online version of this article includes the following figure supplement(s) for figure 3:

**Figure supplement 1.** Irradiation reveals that *pou4−2⁺* cells include progenitors.

by RNA-seq or the *S. mediterranea* scRNA-seq database (*Figures 2–3*; see *Supplementary file 2* for the list of genes). Four of the selected genes, *cadherin-23* (*cdh-23*), *Ephrin Receptor 1* (*EphR1*), *lipoxygenase homology domain-1* (*loxhd-1*), and *unconventional myosin VIIA* (*myo7a*) are predicted to encode proteins homologous to those required for the proper function of inner ear hair cells in humans. Mutations in the human homologs of these genes are associated with sensorineural hearing loss (described below). We used WISH and RNAi analyses to determine their spatial expression patterns and assess whether expression was downregulated following *pou4-2* or *soxB1-2* knock-down (*Figure 4A–B*). The expression patterns of *calmodulin-2* (*calm-2*), *loxhd-1*, and *dd_28678* were confined to the stereotypical mechanosensory neuron pattern in the head tip, body periphery, and dorsal ciliated stripe, and were completely depleted in *pou4-2(RNAi)* and *soxB1-2(RNAi)* planarians (*Figure 4A*). *pou4-2* RNAi-mediated loss of *calm2* expression is consistent with observations reported by *King et al., 2024*. Calmodulins are important for ion channel activity and signal transduction, and human *CALM2* mutations are associated with delayed neurodevelopment and epilepsy (*Crotti et al., 2013*). *loxhd1* is predicted to encode a highly conserved stereociliary protein involved in hair cell function, and mutated human *LOXHD1* causes DFNB77, a form of progressive hearing loss (*Grillet et al., 2009*). *NOP2/Sun RNA methyltransferase family member 7* (*nsun-7*) was expressed in fewer cells in the mechanosensory neuron pattern overall, but an additional subset of *nsun-7⁺* cells present in the optic cups was unaffected by *pou4-2* and *soxB1-2* inhibition, in contrast to the clearly reduced *nsun-7* expression in sensory mechanosensory neuron-rich areas (*Figure 4A*). In humans, *NSUN7* activity is required for proper flagella movement and sperm motility, and mutations result in male infertility (*Khosronezhad et al., 2015*). In control and *pou4-2(RNAi)* planarians, expression of the monocarboxylate transporter gene *solute carrier family 16 member 24* (*slc16a-24*) was detected in the auricles. Auricular *slc16a-24* expression was downregulated in *soxB1-2(RNAi)* but not *pou4-2(RNAi)* planarians, while expression in the mechanosensory neurons important for rheosensation was downregulated in both (*Figure 4A*).

Other genes we chose to analyze were not exclusively expressed in the ciliated stripes (*Figure 4B*). Expression of *cadherin-23* (*cdh23*) was highest in the photoreceptors, and low expression was detected in the epidermis. While *cdh23* expression was downregulated in the ciliated stripes, *cdh23⁺* cells in the epidermis and photoreceptors were unchanged after *pou4-2* and *soxB1-2* inhibition. Human *CDH23* is expressed in the sensory epithelium of the inner ear, where it is involved in maintaining the stereocilium organization of hair cells required for sound perception and equilibrioception (*Kazmierczak et al., 2007*), and *CDH23* mutations are known to cause hereditary hearing loss (*Woo et al., 2014*). *Smed-EphR1* (*EphR1*), encoding an ephrin receptor homolog, was also among the differentially expressed genes in *soxB1-2⁺/pou4−2⁺* neurons (*Figure 2C*). The role of Ephrin signaling in axon guidance is well-established and highly conserved; in mammals, the binding of ligand *Efnb2* to receptor *EphA4* is critical to the differentiation and patterning of hair and support cells on the cochlear sensory epithelium (*Defourny et al., 2019*) and for targeting and innervating auditory projections to hair cells (*Defourny et al., 2013*). In humans, mutations in *EPHA4* and disruption of ephrin ligand binding are associated with sensorineural hearing loss (*Lévy et al., 2018*). While mechanosensory neuronal patterned expression of *EphR1* was downregulated after *pou4-2* and *soxB1-2* inhibition, low expression in the brain branches of the ventral cephalic ganglia persisted (*Figure 4B*). *EphR1* expression in the anterior-most region of the head tip was downregulated in *soxB1-2(RNAi)* but was unaffected in *pou4-2(RNAi)* planarians (*Figure 4B*). *gelsolin-2* (*glsn-2*) was highly expressed in the sensory neuron pattern and in the epidermis, where it appeared most abundantly expressed near the body periphery and weakly

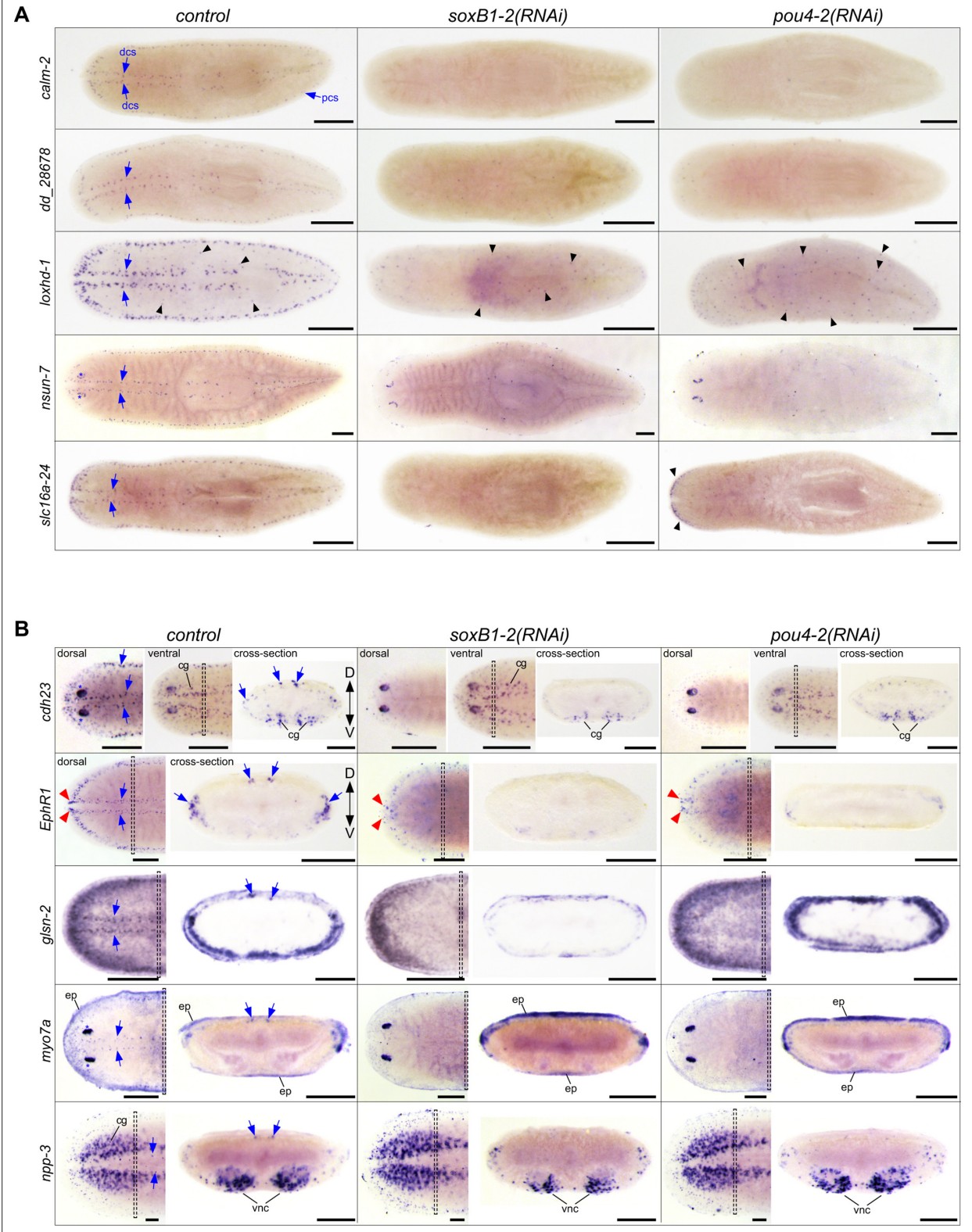

**Figure 4.** Expression analysis of genes co-expressed in *Smed-pou4–2*+cells. (**A**) WISH images of genes predominantly expressed in mechanosensory neuron regions, dorsal and peripheral ciliated stripes (dcs, pcs). WISH post-RNAi revealed reduced expression of mechanosensory neuron-patterned genes (labeled on the left) after *soxB1-2* and *pou4-2* RNAi (labeled on the top). *loxhd-1* was also expressed in a punctate pattern (black arrowheads) that appeared largely unaffected following *pou4-2* RNAi. The RNAi treatments did not affect *nsun-7* expression in the photoreceptors (blue asterisks). (**B**) In

*Figure 4 continued on next page*

*Figure 4 continued*

situ hybridization images from whole-mount and cross-sections of genes expressed in mechanosensory neurons and other cell types. Note reduced expression of genes in the stereotypical mechanosensory neuron regions after *soxB1-2* and *pou4-2* RNAi. The red arrowheads highlight the head tip expression unaffected by *soxB1-2* and *pou4-2* knockdown in *EphR1*-labeled cells. The insets show the corresponding cross-section of the worm. Anterior to the left. Blue arrows mark ciliated stripe cell regions. Dashed boxes denote cross-section regions. Abbreviations: cephalic ganglia (cg), dorsal ciliated stripe (dcs), dorsal and ventral peripheral stripes (pcs), epidermis (ep), ventral nerve cords (vnc). Scale bars = 200 µm for intact animals and 100 µm for cross-sections; n ≥3 worms tested with all samples displaying similar expression patterns.

expressed in the medial dorsal surface (*Figure 5A*). Gelsolins' roles in nervous system development (*Mazur et al., 2016*) and as modulators of ciliogenesis and cilium length are evolutionarily conserved (*Kim et al., 2010*). Mechanosensory *glsn-2* expression was downregulated in both *pou4-2(RNAi)* and *soxB1-2(RNAi)* planarians. Epidermal expression of *glsn-2* was also reduced in *soxB1-2(RNAi)* animals, consistent with its broader role in ciliated epidermal cell maintenance (*Ross et al., 2018*).

In the dorsal ciliated stripe, body periphery, and head tip, low expression of *unconventional myosin VIIA* (*myo7a*) was detected and depleted in *pou4-2(RNAi)* and *soxB1-2(RNAi)* planarians (*Figure 4B*). Additionally, *myo7a* was highly expressed in the photoreceptors, and this expression remained in *pou4-2(RNAi)* and *soxB1-2(RNAi)* planarians. The low expression of *myo7a* detected beneath the epidermis was also unaffected following *pou4-2* and *soxB1-2* inhibition. Human *MYO7A* is important in stereocilium organization, differentiation, and signal transduction of inner ear hair cells (*Jaijo et al., 2007*). Defective *MYO7A* and *CDH23*, to a lesser extent, cause Usher Syndrome Type 1B (USH1B), which is characterized by deafness and reduced vestibular function (*Roux et al., 2006*). *neuropeptide precursor-3* (*npp-3*) was highly expressed in the cephalic ganglia, ventral nerve cords, pharynx, and at lower levels in the parenchyma. A small subset of *npp-3*⁺ cells in the dorsal ciliated stripe was depleted in *pou4-2(RNAi)* and *soxB1-2(RNAi)* planarians (*Figure 4B*). Despite the presence of *pou4-2*⁺ cells in the ventral nerve cords (*Figure 1B*), *pou4-2* inhibition appeared not to affect *npp-3* expression in that region (*Figure 5B*).

## Expression of *Smed-pou4-2* is required for mechanosensory neuron regeneration and function

Given that *pou4-2* expression is reduced in *soxB1-2(RNAi)* planarians (*Figure 2D*) and that *pou4-2(RNAi)* animals show downregulation of mechanosensory and cilia-related genes (*Figures 3–4*), we reasoned that a subset of *pou4-2*⁺ cells represents terminally differentiated ciliated sensory neurons. Moreover, the requirement of *soxB1-2* for maintaining ciliated epidermal and sensory neuron populations supports the idea that *pou4-2* functions as a terminal selector in planarians. To assess the role of *pou4-2* in the dorsal and peripheral ciliated stripes, we first immunostained control and RNAi-treated planarians with anti-Acetylated-Tubulin to mark cilia. Compared to the controls, *pou4-2(RNAi)* planarians showed a disruption of the stereotypical banded pattern with decreased cilia labeling along the dorsal ciliated stripe, while ciliated lawns on the dorsal and ventral surfaces remained unchanged (*Figure 5A–B*). To investigate the role of *pou4-2* in mechanosensory function, we used a semi-automated behavioral assay to evaluate the mechanosensory response to vibration stimulation (*Ross et al., 2024*; *Figure 5C*). Knockdown of *pou4-2* led to a significant reduction in vibration-induced contraction behavior in both intact and regenerating animals (*Figure 5D–E*). Thus, we conclude *pou4-2* is downstream of *soxB1-2* and is necessary for maintaining and regenerating ciliated mechanosensory neurons in the rheosensory organ (*Figure 5F*).

## Discussion
### *Smed-pou4-2* plays a key role in the regulation of sensory system differentiation

The interplay between lineage-specifying transcription factors and their respective gene regulatory networks coordinates precise developmental processes and stem cell fate decisions. POU4 transcription factors are conserved terminal selectors of sensory neuron fate, but the role of *Pou4* has not been extensively characterized in regeneration. This study's objective was to elucidate the role of *Smed-pou4-2* (*pou4-2*) in planarian sensory neuron regeneration and to identify gene regulatory network components responsible for maintaining mechanosensory neuron function. *pou4-2*⁺ cells

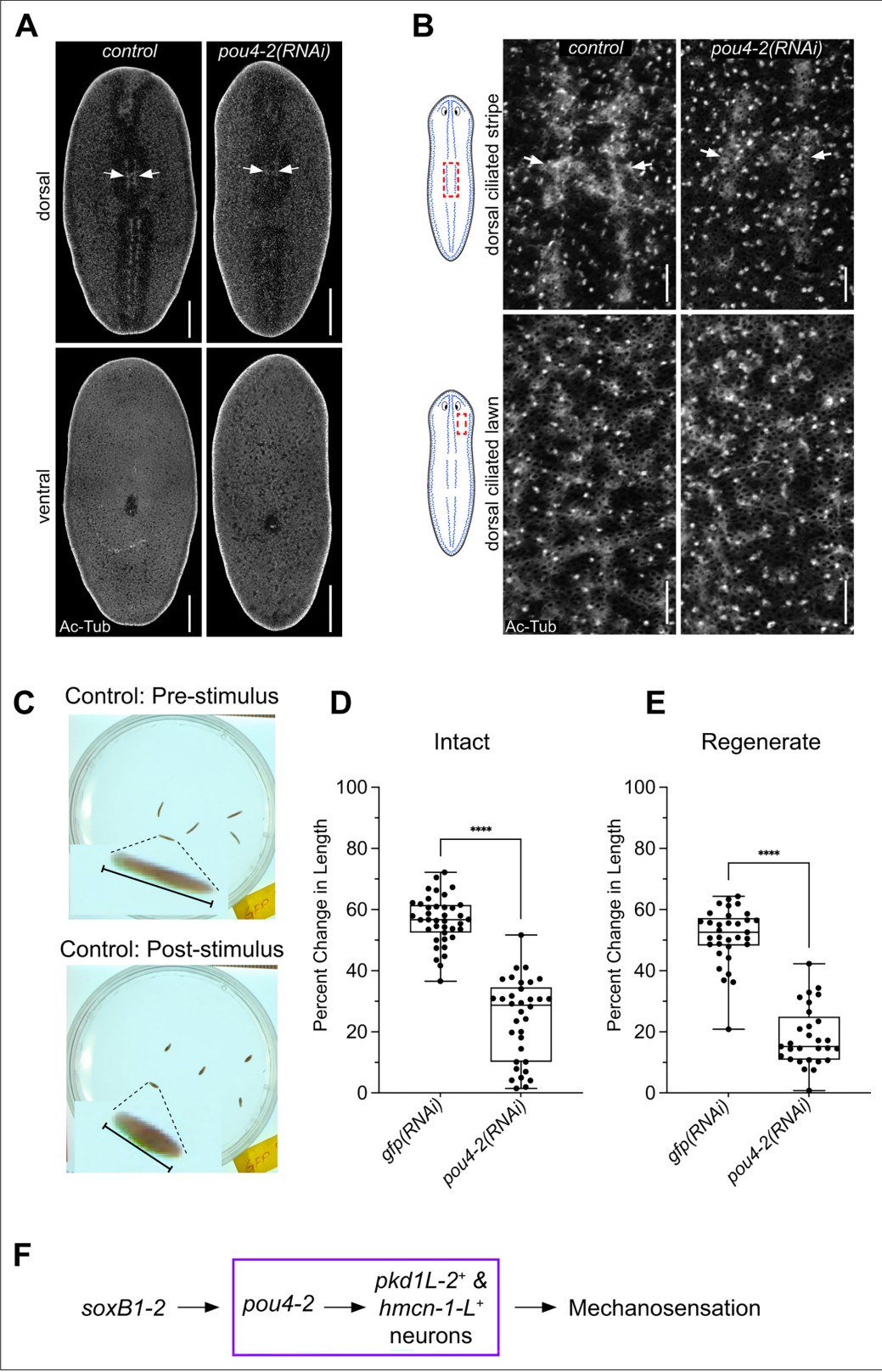

**Figure 5.** *pou4-2* expression is required for mechanosensory neuron regeneration and function. (**A**) Acetylated-tubulin staining shows decreased cilia signal in the dorsal ciliated stripe of *pou4-2(RNAi)* animals. Scale bars = 200 μm, n = 4 worms stained for each of the control and experimental groups. (**B**) Higher magnification confirms stripe reduction; epidermal and ventral cilia are unaffected. Scale bars = 25 μm. (**C**) Vibration response assay

*Figure 5 continued on next page*

*Figure 5 continued*

demonstrating body contractions in wild-type animals following tapping stimulus. (**D, E**) Quantification of vibration response assay data shows significantly reduced contraction responses in both intact (**D**) and regenerate (**E**) *pou4-2(RNAi)* animals. Data in D and E are represented as mean ± SD; n ≥25 worms for each experimental group. ****p<0.0001, Student's t-test. (**F**) Model: *pou4-2* acts downstream of *soxB1-2* in regulating mechanosensory neuron differentiation.

include ciliated mechanosensory neurons that allow planarians to detect water currents and vibrations (rheosensation), a function similar to *POU4F3* in hair cells of the mammalian inner ear sensory epithelium responsible for auditory perception and equilibrioception. We showed that *pou4-2* regulates the expression of genes homologous to those involved in stereocilium organization, cell adhesion, and nervous system development in other organisms. Several of these human homologs are implicated in sensorineural hearing loss. These findings shed light on the molecular mechanisms underlying planarian regeneration and provide insight into the conserved role of POU4 transcription factors in sensory neuron development across divergent species.

Differential expression analysis following *pou4-2* knockdown revealed several genes homologous to those with known roles in mechanosensation in other organisms, including *loxhd-1*, *cdh23*, and *myo7a*. In vertebrate systems, mutations in these genes disrupt mechanosensory function and contribute to hearing loss. For example, a recent study has demonstrated that *Loxhd-1* mutation in the inner hair cell does not affect the structural integrity of the hair cell bundle but rather prevents the activation of MET (mechanoelectrical transducer) channels, thereby contributing to progressive hearing loss in mice (*Trouillet et al., 2021*). Other interesting candidates included a planarian homolog of Ephrin receptors, *EphR1,* which was also expressed in *soxB1-2$^+$/pou4−2$^+$*neurons and was downstream of these two transcription factors. This observation highlights the potential for planarians to serve as a discovery platform for conserved regulators of sensory neuron patterning. We discovered that *EphR1* is required for patterning of mechanosensory neurons in *S. mediterranea* (*McCubbin, 2022*; *Warner, 2024*), which led us to examine in detail how Ephrin signaling genes contribute to neural patterning in planarians (unpublished observations). Thus, as we demonstrated with *soxB1-2* (*Ross et al., 2024*; *Ross et al., 2018*), planarians are also useful to analyze *pou4-2*-regulated genes and their roles in cell differentiation or mechanosensation. It will be important to perform a comparative analysis of Pou4-regulated genes gleaned from other animals, like sea anemones or vertebrates like birds and fish, which can regenerate hair cells or the lateral line, respectively (*Chen et al., 2019*; *Tournière et al., 2020*; *Zhao et al., 2020*). However, we found that our RNA-seq experimental design had limitations in detecting *pou4-2*-regulated transcripts, likely due to the use of systemic RNAi and bulk RNA extraction from whole animals. Before leveraging existing transcriptomic data on Pou4 homologs or hair cell or lateral line regeneration (*Jiang et al., 2014*; *Ku et al., 2014*; *Tournière et al., 2020*) for orthologous comparisons in other species to test to what extent the Pou4 gene regulatory network is conserved among these widely divergent animals, we have designed new experiments to enrich for *pou4-2*-expressing planarian tissues and have performed RNA-seq experiments producing a larger differentially expressed gene set (unpublished). In addition, ATAC-seq experiments could be performed to examine how *pou4-2* activity affects chromatin architecture. These future genomic experiments should build upon this work and improve the resolution of the *pou4-2* gene regulatory network implicated in sensory neuron regeneration.

*pou4-2* function could encompass additional roles other than the ones identified in this study. In addition to the *pkd1L-2$^+$* and *hmcn-1-L$^+$* populations present in the rheosensory organ, there is a population of *pou4-2$^+$* cells in the ventral nerve cords and cephalic ganglia, which are not regulated by SoxB1-2 (*Figure 1A*). Although the ventral nerve cords are populated by *npp-3$^+$* cells, Pou4-2 activity does not regulate *npp-3* expression in this region as it does in the rheosensory organ (*Figure 4B*), and it has not been confirmed whether any cells in the ventral nerve cords are *pou4-2$^+$/npp-3$^+$* co-expressing cells. While this study did not define the function of this *pou4-2$^+$* central nervous system population, it may be possible to mine new scRNA-seq datasets to uncover transcription factors, such as in *King et al., 2024*, to predict the identities of *pou4−2$^+$* cells negative for expression of sensory neuron markers like *pkd1L-2* and *hmcn-1-L*.

Our results indicate that the regulatory relationship between *pou4* and *atonal* observed in other species does not appear to be conserved in planarians. The expression patterns of planarian *atonal*

genes indicated that they represent completely different cell populations from *pou4-2*-regulated mechanosensory neurons. Although several *pou4-2*-regulated genes are also expressed outside the rheosensory organ, *pou4-2* appears to primarily regulate gene expression in mechanosensory neuron-enriched regions, consistent with prior observations in *pou4-2(RNAi)* planarians (*Elliott, 2016*; *King et al., 2024*).

### *Smed-pou4-2* is required for the regeneration of mechanosensory function

RNAi of *pou4-2* substantially reduced the expression of the terminal mechanosensory markers *pkd1L-2* and *hmcn-1-L* in intact animals, whereas *pou4-2(RNAi)* regenerate animals had a markedly reduced, dispersed expression of these genes at the head tips and peripheral ciliated stripes (*Figure 3B*). These findings raise the possibility that following injury, *pou4-2*+ progenitors or committed precursors may still undergo limited differentiation into sensory neurons in response to injury and polarity cues. This is consistent with previous observations in animals treated with hydroxyurea (HU) to block stem cell progression through S-phase. In these experiments, *Smed-APC-1(RNAi)* and *Smed-ptc(RNAi)* HU-treated animals were still able to regenerate neurons (*Evans et al., 2011*). It is also consistent with the hypothesis that many planarian stem cells are already specialized (*Raz et al., 2021*) and may have been unaffected by our RNAi treatment scheme. Nevertheless, both *pou4-2(RNAi)* intact animals and regenerates exhibited a significant reduction in mechanosensory responses compared to controls (*Figure 5D–E*). Moreover, the late re-expression of *pou4-2* during regeneration, combined with the more rapid depletion of *pou4-2*+ cells lacking *pkd1L-2* and *hmcn-1-L* expression compared to *pou4-2*+ cells co-expressing these terminal markers in irradiated animals, supports the idea that *pou4-2*+ progenitors give rise to terminally differentiated mechanosensory neurons (*Figure 3—figure supplement 1*).

*pou4-2*+ cells in the dorsal head tip and peripheral and dorsal ciliated stripes (the planarian rheosensory organ) are downstream of SoxB1-2 activity, and RNAi and WISH experiments demonstrated that *pou4-2* expression is necessary for maintaining the functional properties of mechanosensory neurons (*Figure 2D–E*). However, the planarian rheosensory organ is composed of both ciliated mechanosensory neuronal and epidermal populations. Since *soxB1-2* is a SoxB1 family transcription factor involved in ectodermal lineage specification, RNAi of *soxB1-2* resulted in the loss of both ciliated sensory neuronal and epidermal populations in the rheosensory organ of planarian; thus, the dorsal ciliated stripe observed in acetylated tubulin labeling disappeared entirely (*Ross et al., 2018*). In contrast, in *pou4-2(RNAi)* animals, labeling of ciliated mechanosensory neurons was markedly reduced, but epidermal ciliated cells within the stripe appeared largely unaffected (*Figure 5A–B*). These findings support the conclusion that *pou4-2*+ cells represent a subset of mechanosensory neurons within the rheosensory organ and that *pou4-2* is not required for the maintenance of ciliated epidermal cells, unlike *soxB1-2*, which has a broader role in ectodermal lineage regulation.

### Concluding remarks

Despite molecular evidence indicating planarians possess ciliated mechanoreceptors sharing homology with mechanoreceptor function and development in other organisms, we have yet to fully resolve the cellular morphologies of the collection of cells comprising the ciliated stripes. Although it remains uncertain whether ciliated mechanoreceptors are products of convergent evolution or share a common cellular ancestry (*Manley and Ladher, 2008*), the role of Pou4 appears to represent a critical component of an adaptable gene regulatory network that has been co-opted to manufacture mechanoreceptors in distinct cell types. In addition, the mechanisms that specify *pou4*+ progenitors in response to local cues remain unclear. Studies have implicated Notch signaling as the likely culprit (*Elliott, 2016*), and recent studies in *Schmidtea mediterranea* have elegantly demonstrated that Notch signaling plays a role in patterning neurons and glial cells (*Scimone et al., 2025*). Together, our findings demonstrate that *pou4-2* is critical for the maintenance, regeneration, and function of ciliated mechanosensory neurons in *S. mediterranea*. Future studies aimed at identifying additional *pou4-2* target genes and defining how *pou4-2* influences chromatin accessibility will provide deeper insight into the transcriptional regulation of mechanosensory regeneration in planarians.

## Materials and methods

### Planarian culture

Asexual clonal line CIW4 of *S. mediterranea* was maintained in 1 x Montjuïc salts (1.6 mM NaCl, 1.0 mM CaCl$_2$, 1.0 mM MgSO$_4$, 0.1 mM MgCl$_2$, 0.1 mM KCl, and 1.2 mM NaHCO$_3$) in the dark at 20 °C and fed weekly with pureed calf liver (*Merryman et al., 2018*). Planarians 3–5 mm in length were starved for 1 week before experimentation unless specified otherwise.

### Gene identification and cloning

Sequences were obtained from an EST library (*Zayas et al., 2005*), cloned using gene-specific primers, or synthesized as eBlocks (IDT) and inserted into pPR-T4P (*Liu et al., 2013*) or pJC53.2 (*Collins et al., 2010*) vectors through ligation-independent cloning. Primer sequences, eBlock sequences, and EST clone accession numbers are listed in *Supplementary file 5*.

### In situ hybridization

Riboprobes were synthesized using an in vitro transcription reaction from DNA templates with digoxigenin or fluorescein-labeled NTPs, and whole-mount in situ hybridizations were performed as previously described (*King and Newmark, 2013*) in an InsituPro automated liquid handling robot (CEM Corporation, Matthews, NC). Briefly, samples were incubated with anti-Digoxigenin-AP (1:2000, Roche) for chromogenic detection, and the signals were subsequently developed with NBT/BCIP in AP buffer. For double fluorescent in situ hybridizations (dFISH), samples were incubated for 16 hr at 4 °C with anti-DIG-AP and anti-FITC-POD (1:250, Roche). Peroxidase-conjugates were detected with tyramide signal amplification (TSA) as outlined previously in *Brown and Pearson, 2015*, and Fast Blue development was utilized for AP-driven reaction detection (*Lauter et al., 2011*).

### scRNA sequencing data analysis

To infer the gene expression profiles of *pou4–2*$^+$ cells, we analyzed publicly available scRNA-seq data from *S. mediterranea* [GSE111764] (*Fincher et al., 2018*). From the whole-body data (50,562 cells) and the brain data (7,766 cells), we extracted putative neuronal cells, defined as those expressing at least one of the following transcripts: *synapsin* (dd_Smed_v4_3135_0_1), *synaptotagmin* (dd_Smed_v4_4222_0_1, dd_Smed_v4_6730_0_1, dd_Smed_v4_6920_0_1), and *synaptosome associated protein 25* (dd_Smed_v4_13079_0_1, dd_Smed_v4_13255_0_1, or dd_Smed_v4_3977_0_1). This yielded 20,557 cells of which 1,427 expressed *soxB1-2* (dd_Smed_v4_8104_0_1). Expression data were scaled and transformed using standard functions of the R library Seurat (*Satija et al., 2015*). We performed dimensionality reduction using UMAP and clustering using the Leiden algorithm (*Traag et al., 2019*). Since the UMAP projection suggested communities of disparate size, we used the Leiden algorithm in two steps – once with a smaller resolution to capture the large-scale structure (10 clusters) and then selectively with a higher resolution to resolve smaller communities, for a total of 19 distinct clusters. We characterized each cluster by finding differentially expressed genes (markers) using the bootstrap method of Pollard and van der Laan (*Pollard and Laan, 2005*) to calculate the *z*-score for every gene between cells in each cluster relative to cells outside the cluster (*Supplementary file 1*). The *z*-scores are then assessed for significance using the empirical Bayes method of *Efron, 2008*. The result is a posterior error probability *lfdr* assigned to each gene.

### RNA interference (RNAi)

Bacterially expressed dsRNA was prepared by cloning gene-specific fragments into pPR-T4P or pJC.53.2 vectors, then transforming them into HT115 *E. coli*. Briefly, cultures grown overnight in LB with appropriate antibiotics were diluted in 40 mL 2xYT and incubated at 37 °C and shaking at 225 rpm. Once cultures reached 0.6–0.8 OD600, 1 mM IPTG was added to induce dsRNA synthesis, and incubation continued for 2 hr. Cultures were then pelleted at 3,000 x *gg* for 10 min at 4 °C, resuspended in 8 mL LB, and aliquoted into eight microcentrifuge tubes. Resuspended cultures were pelleted at 11,000 x *gg* for 5 min at 4 °C, aspirated, and stored at –80 °C. For RNAi feeding, bacterial pellets were mixed with liver puree as previously described (*Gurley et al., 2008*). Animals were fed eight times over 4 weeks, with *gfp* dsRNA used as a negative control. For stainings, intact animals were fixed 10 days after the 8th feed, and regenerated animals were pre-pharyngeally amputated one

day after the 8th feed and fixed after 10 days of regeneration. RNAi feedings for the behavioral assays were performed using in vitro transcribed dsRNA mixed with pureed liver and agarose, as described in *Ross et al., 2024*. For intact animals, assays were performed three days after the 8th feed. For regenerates, animals were amputated 24 hr after the 8th feed and tested for behavioral defects 14–17 days of regeneration. The total numbers of animals for RNAi experiments are summarized in *Supplementary file 6*.

## RNA sequencing

Three biological replicates were obtained, each consisting of four worms of approximately four mm length at the start of the experiment, which were starved for one week prior to the start of RNAi feeding. Worms were fed bacterially expressed dsRNA three times on days 0, 3, and 7, and RNA was extracted and purified on day 12 (*Allen et al., 2021*). 500 ng of total RNA was used for the RNA-seq library preparation and sequencing at MedGenome, Inc (Foster City, CA). The Poly-A-containing mRNA molecules were purified using poly-T oligo attached magnetic beads, and then the mRNA was converted to cDNA using Illumina TruSeq stranded mRNA kit (20020595) according to the manufacturer's protocol. Libraries were sequenced for 100 cycles to a depth of 30 million paired reads using Illumina NovaSeq 6000 (Illumina, San Diego, CA). The following quality control steps were performed on the fastq files: Base quality score distribution, Sequence quality score distribution, Average base content per read, GC distribution in the reads, distribution of over-represented sequences, and adapter trimming. Based on the quality report of fastq files, sequences were trimmed wherever necessary to retain only high-quality sequences for further analysis. In addition, the low-quality sequence reads are excluded from the analysis. Data quality check was performed using FastQC (v0.11.8). The adapter trimming was performed using the fastq-mcf program (v1.05) and cutadapt (v2.5; *Martin, 2011*). Transcriptome alignment was performed using RSEM (version RSEM v1.3.1; *Li and Dewey, 2011*) against the dd_Smed_v6 transcriptome (*Rozanski et al., 2019*) to build Bowtie transcriptome indexes using *rsem-prepare-reference*, then used *rsem-calculate-expression* for aligning and expression calculation. Differential expression analysis was performed using DESeq2 (R Bioconductor package; *Anders and Huber, 2010*) with default parameters, and then differentially reduced genes were defined as those having a fold-change of <1.4 and p-adjusted value <0.1 (see *Supplementary file 3*). The RNA sequencing data have been deposited in NCBI under BioProject accession PRJNA1258257. For Gene Ontology (GO) annotation and over-presentation analysis, the unique set of *pou4-2(RNAi)* down-regulated transcripts was compared to the human proteome (BLASTX against the Swiss-Prot *Homo sapiens* proteome, cutoff e-value<1e$^{-3}$). Human UniProt IDs were used for enrichment analysis using Fisher's Exact tests with FDR multiple test correction (FDR <0.05) in http://geneontology.org/. GO results are reported in *Supplementary file 4*.

## Immunohistochemistry

Animals were sacrificed in ice-cold 2% HCl for 30 s, followed by incubation in Carnoy's fixative (6 parts ethanol: 3 parts CHCl3: 1 part glacial acetic acid) for 2 hr at 4 °C (*Forsthoefel et al., 2018*), followed by a dehydration step with 100% methanol for 1 hr at 4 °C. The animals were bleached overnight under a lamp with 6% $H_2O_2$ in methanol and then rehydrated in 75%, 50%, and 25% methanol-PBSTx, followed by two 5-min PBSTx washes. PBSTb (1% BSA in PBSTx) was used for blocking at room temperature for 2 hr. Primary antibody labeling was carried out with mouse anti-Acetylated Tubulin (Sigma-Aldrich, St. Louis, MO) diluted in PBSTb (1:1000) overnight at 4 °C. Six 1 hr PBSTx washes followed by 1 hr of PBSTb blocking were performed before anti-mouse-HRP (1:1000, Cell Signaling) incubation overnight at 4 °C. After six 1 hr PBSTx washes, acetylated tubulin was detected through TSA development as described in *Brown and Pearson, 2015* with the following exceptions: no 4-IPBA or dextran sulfate was added to the TSA reaction buffer, Cy3-tyramide was diluted 1:250, and development took place for a total of 20 min.

## Mechanosensation (vibration) assay

Analysis of the planarian's ability to detect a vibration stimulus was conducted essentially as described in *Ross et al., 2024*. Briefly, groups of five control *gfp(RNAi)* or *pou4-2(RNAi)* planarians were added to a 100×15 mm petri dish containing 40 ml of 1 x Montjuïc salts that was placed inside a dish lid that was mounted to a cold LED lighted board using clear silicone paste and observed until gliding

normally. Then, an Arduino-controlled arm delivered five taps at a rate of one tap every 75 ms to the side of the dish. Experimental runs were recorded on a Basler Ace 2 Pro ac1440-220uc camera connected to a PC running Basler's pylon Viewer 64-bit version 6.3.0 software at a frame rate of 10 frames/sec and a frame size of 1440×1080 pixels. The video frames were analyzed in Fiji (ImageJ2 version 2.9.0; *Schindelin et al., 2012*), using the line tool to measure the longest pre-stimulus gliding length and the length of the worms following the stimulus. The percent change in length was calculated as $[(\text{Length}_{Prestimulus} - \text{Length}_{Poststimulus})/\text{Length}_{Prestimulus}] \times 100$. Statistical analysis and graph generation were performed in GraphPad Prism 9 (GraphPad Software, Boston, MA). One-way ANOVA analyses were performed and corrected using Dunnett's correction. All means were compared to the control group, and statistical significance was accepted at values of $p < 0.01$.

## X-ray irradiation

Starved animals (3–4 mm) were irradiated with 100 Gy of X-rays (130 kV, 5 mA, 8.4 Gy/min) for approximately 12 min using a CellRad irradiator (Precision X-Ray, Madison, CT).

## Imaging

Brightfield images were acquired with a Leica DFC450 camera and M205 stereomicroscope. Animals processed for fluorescent in situ hybridization and immunohistochemistry were mounted in Vectashield diluted 1:1 in 80% glycerol. Fluorescent images were acquired using a Zeiss AxioZoom equipped with an Apotome using Zen Pro version. High-magnification acetylated tubulin images were acquired using a Zeiss Axio Observer Inverted Microscope equipped with an AV4 Mod Apotome using AxioVision v4.6 (Carl Zeiss Microscopy, LLC, White Plains, NY).

## Cell counting and quantification

For co-labeling experiments between *pou4-2*-regulated genes with *pkd1L-2* or *hmcn-1-L*, maximum intensity projections of stacked fluorescent images were acquired with a depth of 12 μm. For *pou4-2* co-labeling with *pkd1L-2* and *hmcn-1-L* in irradiated planarians, maximum intensity projections of stacked fluorescent images were acquired with a depth of 24 μm. The regions quantified spanned the width of the sensory neuron expression pattern in the head tip and 500 μm anterior to the head tip along the length of the dorsal ciliated stripe. Cells were manually counted using Zeiss ZEN lite v3.3. Cell quantification was represented as the number of positive cells per mm², and graphs were made using GraphPad Prism (GraphPad Software, Boston, MA). Three to six biological replicates per group were used to quantify co-labeling. Cell quantification details and the total number of cells counted for *Figure 3B*, *Figure 3—figure supplement 1A–B* are summarized in *Supplementary files 7 and 8*, respectively.

## Acknowledgements

This work was supported by a California Institute for Regenerative Medicine (CIRM) postdoctoral fellowship (EDUC4-12813) to MAA, NIH R01GM135657 to RMZ, and NSF IOS Grants 557448 and 1938531 to RWZ. We thank Dr. Victoria Hurless for initiating the cloning and analysis of *Smed-pou4-2*, Dr. John Allen for assistance with RNA extractions, and Dr. Peter Reddien for generously sharing annotated scRNA-seq data files.

## Additional information

### Funding

| Funder | Grant reference number | Author |
| --- | --- | --- |
| California Institute for Regenerative Medicine | EDUC4-12813 | Mohammad A Auwal |
| National Institutes of Health | NIH R01GM135657 | Ricardo M Zayas |

| Funder | Grant reference number | Author |
|---|---|---|
| U.S. National Science Foundation | IOS 557448 | Robert W Zeller |
| U.S. National Science Foundation | IOS 1938531 | Robert W Zeller |

The funders had no role in study design, data collection and interpretation, or the decision to submit the work for publication.

### Author contributions

Ryan A McCubbin, Formal analysis, Investigation, Writing – original draft; Mohammad A Auwal, Formal analysis, Investigation, Writing – original draft, Writing – review and editing; Shengzhou Wang, Sarai Alvarez Zepeda, Investigation; Roman Sasik, Software, Formal analysis, Investigation; Robert W Zeller, Conceptualization, Supervision; Kelly G Ross, Formal analysis, Supervision, Writing – original draft, Project administration, Writing – review and editing; Ricardo M Zayas, Conceptualization, Supervision, Funding acquisition, Writing – original draft, Project administration, Writing – review and editing

### Author ORCIDs

Sarai Alvarez Zepeda ⓘ https://orcid.org/0009-0003-8740-6280
Kelly G Ross ⓘ https://orcid.org/0000-0002-5940-8778
Ricardo M Zayas ⓘ https://orcid.org/0000-0002-6272-0519

Reviewer #1 (Public review): https://doi.org/10.7554/eLife.107718.3.sa1
Reviewer #2 (Public review): https://doi.org/10.7554/eLife.107718.3.sa2
Author response https://doi.org/10.7554/eLife.107718.3.sa3

## Additional files

### Supplementary files

Supplementary file 1. Gene IDs for transcripts enriched in scRNA-seq *soxB1−2*+neuronal clusters shown in *Figure 2A*.

Supplementary file 2. Gene IDs for transcripts shown in *Figure 2C* heatmap.

Supplementary file 3. Differentially expressed genes in *pou4-2* RNAi planarians plotted in *Figure 3A*.

Supplementary file 4. Gene Ontology analysis of downregulated genes in *pou4-2* RNAi planarians.

Supplementary file 5. Primer sequences, eBlock sequences, and accession numbers for genes analyzed in this study.

Supplementary file 6. Replicate numbers of animals used for RNAi experiments reported in this study.

Supplementary file 7. Cell quantification details and total number of cells counted for *Figure 3B*.

Supplementary file 8. Cell quantification details and total number of cells counted for *Figure 3—figure supplement 1A–B*.

MDAR checklist

### Data availability

The RNA sequencing data have been deposited in NCBI under BioProject accession PRJNA1258257.

The following dataset was generated:

| Author(s) | Year | Dataset title | Dataset URL | Database and Identifier |
|---|---|---|---|---|
| McCubbin RA, Auwal MA, Wang S, Alvarez Zepeda S, Roman S, Zeller RW, Ross KG, Zayas RM | 2025 | Transcriptional profiling of Smed-pou4-2 RNAi planarians | https://www.ncbi.nlm.nih.gov/bioproject/PRJNA1258257 | NCBI BioProject, PRJNA1258257 |

The following previously published dataset was used:

| Author(s) | Year | Dataset title | Dataset URL | Database and Identifier |
|---|---|---|---|---|
| Fincher CT, Wurtzel O, de Hoog T, Kravarik KM, Reddien PW | 2018 | Cell type transcriptome atlas for the planarian *Schmidtea mediterranea* | https://www.ncbi.nlm.nih.gov/geo/query/acc.cgi?acc=GSE111764 | NCBI Gene Expression Omnibus, GSE111764 |

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
