## [Editor Report · eLife Assessment]

This is a **valuable** study that explores the role of the conserved transcription factor POU4-2 in the maintenance, regeneration, and function of planarian mechanosensory neurons. The authors present **convincing** evidence provided by gene expression and functional studies to demonstrate that POU4-2 is required for the maintenance and regeneration of mechanosensory neurons and mechanosensory function in planarians. Furthermore, the authors identify conserved genes associated with human auditory and rheosensory neurons as potential targets of this transcription factor.

---

## [Referee Report · Reviewer #1 (Public review)]

Summary:

In this manuscript, the authors explore the role of the conserved transcription factor POU4-2 in planarian maintenance and regeneration of mechanosensory neurons. The authors explore the role of this transcription factor and identify potential targets of this transcription factor. Importantly, many genes discovered in this work are deeply conserved, with roles in mechanosensation and hearing, indicating that planarians may be a useful model with which to study the roles of these key molecules. This work is important within the field of regenerative neurobiology, but also impactful for those studying evolution of the machinery that is important for human hearing.

Strengths:

The paper is rigorous and thorough, with convincing support for the conclusions of the work.

---

## [Referee Report · Reviewer #2 (Public review)]

Summary:

In this manuscript, the authors investigate the role of the transcription factor Smed-pou4-2 in the maintenance, regeneration and function of mechanosensory neurons in the freshwater planarian *Schmidtea mediterranea*. First, they characterize the expression of pou4-2 in mechanosensory neurons during both homeostasis and regeneration, and examine how its expression is affected by the knockdown of soxB1, 2, a previously identified transcription factor essential for the maintenance and regeneration of these neurons. Second, the authors assess whether pou4-2 is functionally required for the maintenance and regeneration of mechanosensory neurons.

Strengths:

The study provides some new insights into the regulatory role of pou4-2 in the differentiation, maintenance, and regeneration of ciliated mechanosensory neurons in planarians.

---

## [Author Response]

The following is the authors’ response to the original reviews

**Reviewer #1 (Public review):**
Summary:In this manuscript, the authors explore the role of the conserved transcription factor POU4-2 in planarian maintenance and regeneration of mechanosensory neurons. The authors explore the role of this transcription factor and identify potential targets of this transcription factor. Importantly, many genes discovered in this work are deeply conserved, with roles in mechanosensation and hearing, indicating that planarians may be a useful model with which to study the roles of these key molecules. This work is important within the field of regenerative neurobiology, but also impactful for those studying the evolution of the machinery that is important for human hearing.Strengths:The paper is rigorous and thorough, with convincing support for the conclusions of the work.Weaknesses:Weaknesses are relatively minor and could be addressed with additional experiments or changes in writing.
**Reviewer #2 (Public review):**
Summary:In this manuscript, the authors investigate the role of the transcription factor Smed-pou4-2 in the maintenance, regeneration, and function of mechanosensory neurons in the freshwater planarian *Schmidtea mediterranea*. First, they characterize the expression of pou4-2 in mechanosensory neurons during both homeostasis and regeneration, and examine how its expression is affected by the knockdown of soxB1, 2, a previously identified transcription factor essential for the maintenance and regeneration of these neurons. Second, the authors assess whether pou4-2 is functionally required for the maintenance and regeneration of mechanosensory neurons.Strengths:The study provides some new insights into the regulatory role of pou4-2 in the differentiation, maintenance, and regeneration of ciliated mechanosensory neurons in planarians.Weaknesses:The overall scope is relatively limited. The manuscript lacks clear organization, and many of the conclusions would benefit from additional experiments and more rigorous quantification to enhance their strength and impact.
**Reviewing Editor Comments:**
(1) Quantification of pou4-2(+) cells that express (or do not express) hmcn-1-L and/or pkd1L-2(-) is a common suggestion amongst reviewers. It is recognized that Ross et al. (2018) showed that pkd1L-2 and hmcn-1L expression is detected in separate cells by double FISH, and the analysis presented in Supplementary Figure S3 is helpful in showing that some cells expressing pou4-2 (magenta) are not labeled by the combined signal of pkd1L-2 and hmcn-1-L riboprobes (green). However, I am not sure that we can conclude that pkd1L-2 and hmcn-1-L are effectively detected when riboprobes are combined in the analysis. Therefore, quantification of labeled cells as proposed by Reviewers 1 and 2 would help.

Combining riboprobes is a standard approach in the field, and we chose this method as a direct way to determine which cells lack expression of both genes. We agree that providing the raw quantification data would be helpful for readers, and we included this data in Supplementary File S7; the file contains the quantification information for this dFISH experiment represented in Supplementary Figure 3.

(2) It may be helpful to comment on changes (or lack of changes) in atoh gene RNA levels in RNAseq analyses of pou4-2 animals. As mentioned by one of the reviewers, in situs that don't show signal are inconclusive in this regard.

We fully agree with both reviewers. Two of the planarian atonal homologs are difficult to detect and produce background signals, which we attempted and previously reported in Cowles et al. Development (2013). We conceived performing reciprocal RNAi/in situ experiments, born out of curiosity given the reported role of atonal in the pou4 cascade in other organisms. However, these exploratory experiments lacked a strong rationale for inclusion, particularly given that pou4-2 and the atonal homologs do not share expression patterns, co-expression, or differential expression in our RNA-seq dataset. Therefore, we decided to omit the atonal in situs following pou4-2 RNAi. We retained the experiments showing that knockdown of the atonal genes does not show robust effects on the mechanosensory neuron pattern, as expected. We thank the reviewing editor and reviewers for pinpointing the concern. We agree that additional experiments, such as qPCR experiments, would be needed. We reasoned that while these additional experiments could be informative, they are unlikely to alter the key conclusions of this study substantially.

(3) There seem to be typos at bottom of Figure 10 and top of page 11 when referencing to Figure 4B (should be to 5B instead): "While mechanosensory neuronal patterned expression of Eph1 was downregulated after pou4-2 and soxB1-2 inhibition, low expression in the brain branches of the ventral cephalic ganglia persisted (Figure 4B)."

Thank you! We have fixed those.

(4) Typo (page 13; kernel?): "...to test to what extent the Pou4 gene regulatory kernel is conserved among these widely divergent animals."

Regulatory kernels are defined as the minimal sets of interacting genes that drive developmental processes and are the core circuits within a gene regulatory network, but we recognize that this might not be as well known, so we have changed the term to “network” for clarity.

**Reviewer #1 (Recommendations for the authors):**
(1) The authors indicate that they are interested in finding out whether POU4-2 is important in the creation of mechanosensory neurons in adulthood as well as in embryogenesis (in other words, whether the mechanism is "reused during adult tissue maintenance and regeneration"). The manuscript clearly shows that planarian POU4 -2 is important in adult neurogenesis in planarians, but there is no evidence presented to show that this is a recapitulation of embryogenesis. Is pou4-2 expressed in the planarian embryo? This might be possible to examine by ISH or through the evaluation of sequencing data that already exists in the literature.

We agree that these statements should be precise. We have clarified when we make comparisons to the role of Pou4 in sensory system development in other organisms versus its role in the adult planarian. We examined its expression using the existing database of embryonic gene expression. Thanks for hinting at this idea. We performed BLAST in Planosphere (Davies et al., 2017) to cross-reference our clone matching dd_Smed_v6_30562_0_1, which is identical to SMED30002016. The embryonic gene expression for SMED30002016 indicates this gene is expressed at the expected stages given prior knowledge of the timing of organ development in *Schmidtea mediterranea* (a positive trend begins at Stage 5, with a marked increase by Stage 6 that remains comparable to the asexual expression levels shown). We thank the reviewer for pointing out this oversight. We have incorporated this result in the paper as a Supplementary Figure and discuss how we can only speculate that it has a similar role as we detect in the adult asexual worms.

(2) Can it be determined whether the punctate pou4-2+ cells outside of the stripes are progenitors or other neural cell types? Are there pou4-2+ neurons that are not mechanosensory cell types? Could there be other roles for POU4-2 in the neurogenesis of other cell types? It might help to show percentages of overlap in Figure 4A and discuss whether the two populations add up to 100% of cells.

These are good questions that arise in part from other statements that need clarification in the text (pointed out by Reviewer 2). We think some of the dorsal pou4-2^+^ might represent progenitor cells undergoing terminal differentiation (see Supplementary Figure 4). We attempted BrdU pulse chase experiments but were not successful in consistently detecting pou4-2 at sufficient levels with our protocol. In response to this helpful comment, we have included this question as a future direction in the revised Discussion. Finally, we have edited our description of the expression pattern. We already pointed out that there are other cells on the ventral side that are not affected when soxB1-2 is knocked down. We attempted to resolve the potential identity of those cells working with existing scRNA-seq data in collaboration with colleagues, but their low abundance made it difficult to distinguish other populations. While we acknowledge this interesting possibility, we have chosen to focus this report on the role of pou4-2 downstream of soxB1-2, as this represents the most well-supported aspect of the dataset and was positively highlighted by both the reviewer and editor.

(3) The authors discuss many genes from their analysis that play conserved roles in mechanosensation and hearing. Were there any conserved genes that came up in the analysis of pou4-2(RNAi) planarians that have not yet been studied in human hearing and neurodevelopment? I am wondering the extent to which planarians could be used as a discovery system for mechanosensory neuron function and development, and discussion of this point might increase the impact of this paper or provide critical rationale for expanding work on planarian mechanosensation.

Indeed, we agree that planarians could be used to identify conserved genes with roles in mechanosensation and have included this point in the Discussion. In this study, we have focused on demonstrating the conservation of gene regulation. While this study was initially based on a graduate thesis project, we have since generated a more comprehensive dataset from isolated heads, which we are currently analyzing. This has been emphasized in the revised Discussion.

Minor:(1) For Figure 6E, the authors could consider showing data along a negative axis to indicate a decrease in length in response to vibration and to more clearly show that this decrease doesn't occur as strongly after pou4-2(RNAi).

We displayed this behavior as the percent change, as this is a standard way to represent this data. As the percent change is a positive value, we represent the data as these positive values.

(2) The authors should consider quantifying the decrease of pou4-2 mRNA after atonal(RNAi) conditions, either by RT-qPCR or cell quantification. Visually, the signal in the stripes after atoh8-2(RNAi) seems lower, particularly in the tail. The punctate pattern outside the stripes may also be decreased after atoh8-1(RNAi). But quantification might strengthen the argument.

We agree with the reviewer and acknowledge that we should have been more cautious in interpreting these results. Those two genes are difficult to detect and did not show specific patterns in Cowles et al. (2013). The reviewer is correct that additional experiments are necessary before reaching conclusions, but we do not think as discussed earlier we do not think new experiments would provide insights for the major conclusions. These experiments were exploratory in nature and tangential to our main conclusions, especially in the absence of reciprocal evidence e.g., shared expression patterns, co-expression, or differential expression in our RNA-seq data. Therefore, we decided to eliminate the atonal in situs following pou4-2 RNAi.

**Reviewer #2 (Recommendations for the authors):**
A. Expression of pou4-2 in ciliated mechanosensory neurons:(1) The conclusion that pou4-2 is expressed in ciliated mechanosensory neurons is primarily based on co-expression analysis using a published single-cell dataset. Although the authors later show that a subset of pou4-2 cells also express pkd1L-2 (Figure 4A), a known marker of ciliated mechanosensory neurons, this finding is not properly quantified. I recommend moving Figure 4A to earlier in the manuscript (e.g., to Figure 2) and expanding the analysis to include additional known markers of this cell type. Proper quantification of the extent of co-localization is necessary to support the claim robustly.

As pointed out by the reviewer, there is substantive evidence from our lab and other reports. King et al. also showed pou4-2 and pkd1L-2 ‘regulation’ by their scRNA-seq data, and this function is conserved in the acoel Hofstenia miamia (Hulett et al., PNAS 2024). Our analysis shows convincing co-localization by scRNA-seq and expression of soxB1-2 and neural markers in the respective populations. Furthermore, we included colocalization of pou4-2 with mechanosensory genes using fluorescence in situ hybridization (Figure 3B, Supplementary Figure 4, and Supplementary File S7). We are confident the data conclusively show pou4-2 regulates pkd1L-2 expression in a subset of mechanosensory neurons. Given the strength of existing observations and previously published data, we believe that additional staining experiments are not essential to support this conclusion.

(2) There appears to be a conceptual inconsistency in the interpretation of pou4-2 expression dynamics. On one hand, the authors suggest that delayed pou4-2 expression indicates a role in late-stage differentiation (p.6). On the other hand, they propose that pou4-2 may be expressed in undifferentiated progenitors to initiate downstream transcriptional programs (p.8). These interpretations should be reconciled. Additionally, claims regarding pou4-2 expression in progenitor populations should be supported by co-localization with established stem cell or progenitor markers, rather than inferred from signal intensity alone.

This is an excellent point, and we agree with the reviewer that this section requires editing. As described in response to Reviewer 1, we attempted BrdU pulse chase experiments but were not successful in consistently detecting pou4-2 at sufficient levels with our protocol. Furthermore, we could not obtain strong signals in double labeling experiments in pou4-2 in situs combined with piwi-1 or PIWI-1 antibodies. We will include those experiments as a future direction and amend our conclusions accordingly.

(3) The expression pattern shown in Figure 1B raises questions about the precise anatomical localization of pou4-2 cells. It is unclear whether these cells reside in the subepidermal plexus or the deeper submuscular plexus, which represent distinct neuronal layers (Ross et al., 2017). The observed signals near the ventral nerve cords could suggest submuscular localization. To clarify this, higher-resolution imaging and co-staining with region-specific neural markers are recommended.

In Ross et al. (2018), we showed that the pkd1L-2^+^ cells are located submuscularly. The pkd1L-2 cells express pou4-2, thus the pou4-2^+^ cells are located in the same location. Based on co-expression data and co-expression with PKD genes, we are confident it is submuscular.

B. The functional requirements of pou4-2 in the maintenance of mechanosensory neurons:(1) To evaluate the functional role of pou4-2 in maintaining mechanosensory neurons, the authors performed whole-animal RNA-seq on pou4-2(RNAi) and control animals, identifying a significant downregulation of genes associated with mechanosensory neuron expression. However, the presentation of these findings is fragmented across Figures 3, 4, and 5. I recommend consolidating the RNA-seq results (Figure 3) and the subsequent validation of downregulated genes (Figures 4 and 5) into a single, cohesive figure. This would improve the logical flow and clarity of the manuscript.

As suggested by the reviewer, we have combined Figures 3 and 4 (new Figure 3), which we believe improves the flow. We decided to keep Figure 5 (new Figure 4) as a standalone because it focuses on the characterization of new genes revealed by RNAseq and scRNA-seq data mining that were not previously reported in Ross et al. 2018 and

1.

(2) In pou4-2(RNAi) animals, pkd1L-2 expression appears to be entirely lost, while hmcn-1-L shows faint expression in scattered peripheral regions. The authors suggest that an extended RNAi treatment might be necessary to fully eliminate hmcn-1-L expression. However, an alternative explanation is that pou4-2 is not essential for maintaining all hmcn-1-L cells, particularly if pou4-2 expression does not fully overlap with that of hmcn-1-L. This possibility should be acknowledged and discussed.

We agree and have acknowledged this point in the revised text.

(3) On page 9, the section title claims that "Smed-pou4-2 regulates genes involved in ciliated cell structure organization, cell adhesion, and nervous system development." While some differentially expressed genes are indeed annotated with these functions based on homology, the manuscript does not provide experimental evidence supporting their roles in these biological processes in planarians. The title should be revised to avoid overstatement, and the limitations of extrapolating a function solely from gene annotation should be acknowledged.

Excellent point. We have edited the text to indicate that the genes were annotated or implicated.

(4) The cilia staining presented in Figure 6B to support the claim that pou4-2 is required for ciliated cell structure organization is unconvincing. Improved imaging and more targeted analysis (e.g., co-labeling with mechanosensory markers) are needed to support this conclusion.

We have addressed this concern by adjusting the language to be more precise and indicate that the stereotypical banded pattern is disrupted with decreased cilia labeling along the dorsal ciliated stripe. Indeed, our conclusion overstated the observations made with the staining and imaging resolution. Thank you.

C. The functional requirements of pou4-2 in the regeneration of mechanosensory neurons:To evaluate the role of pou4-2 in the regeneration of mechanosensory neurons, the authors performed amputations on pou4-2(RNAi) and control(RNAi) animals and assessed the expression of mechanosensory markers (pkd1L-2, hmcn-1-L) alongside a functional assay. However, the results shown in Figure 4B indicate the presence of numerous pkd1L-2 and hmcn-1-L cells in the blastema of pou4-2(RNAi) animals. This observation raises the possibility that pou4-2 may not be essential for the regeneration of these mechanosensory neurons. The authors should address this alternative interpretation.

Our interpretation is that there were very few cells expressing the markers compared to controls. The pattern was predominantly lost, which is consistent with other experiments shown in the paper. However, we have added the additional caveat suggested by the reviewer.

Minor points:(1) On p.8, the authors wrote "every 12 hours post-irradiation". However, this is not consistent with the figure, which only shows 0, 3, 4, 4.5, 5, and 5.5 dpi.

We corrected this. Thank you for catching the mistake!

(2) On p.12, the authors wrote "Analysis of pou4-2 RNAi data revealed differentially expressed genes with known roles in mechanosensory functions, such as loxhd-1, cdh23, and myo7a. Mutations in these genes can cause a loss of mechanosensation/transduction". This is misleading because, to my knowledge, the role of these genes in planarians is unknown. If the authors meant other model systems, they should clearly state this in the text and include proper references.

The reviewer is correct that we are referencing findings from other organisms. We have clarified this point in the revised text. The appropriate references were included and cited in the first version.

(3) On p.7, the authors wrote, "conversely, the expression of atonal genes was unaffected in pou4-2 RNAi-treated regenerates (Supplementary Figure S2B)". However, it is unclear whether the Atoh8-1 and Atoh8-2 signals are real, as the quality of the in situ results is too low to distinguish between real signals and background noise/non-specific staining.

This valid concern was addressed in our response to Reviewer 1. We have adjusted the figure and the text accordingly.

(4) On p.6 the authors wrote "pinpointed time points wherein the pou4-2 transcripts were robustly downregulated". However, the current version of the manuscript does not provide data explaining why Pou4-2 transcripts are robustly downregulated on day 12.

Yes, we determined the appropriate time points using qPCR for all sample extractions. As an example, see the figure for qPCR validation at day 12 showing that pou4-2 and pkd1L2 are down.

**Author response image 1. sa3fig1:** In this graph, samples labeled “G” represent four biological controls of gfp(RNAi) control animals, and samples labeled “P” represent four biological controls of pou4-2(RNAi)animals at day 12 in the RNAi protocol.

(5) On p.13, the authors wrote "collecting RNA from how animals." Is this a typo?

Thanks for catching the typo. It should read “whole” animals. We have corrected this.

(6) On p.14, the authors wrote "but the expression patterns of planarian atonal genes indicated that they represent completely different cell populations from pou4-2-regulated mechanosensory neurons". However, this is unclear from the images, as the in situ staining of Atoh8-1 and Atoh82 are potentially failed stainings.

We agree. We have edited accordingly.